# The Association between Cyclin Dependent Kinase 2 Associated Protein 1 (CDK2AP1) and Molecular Subtypes of Lethal Prostate Cancer

**DOI:** 10.3390/ijms232113326

**Published:** 2022-11-01

**Authors:** Yaser Gamallat, Andrea Bakker, Ealia Khosh Kish, Muhammad Choudhry, Simon Walker, Saood Aldakheel, Sima Seyedi, Kuo-Cheng Huang, Sunita Ghosh, Geoffrey Gotto, Tarek A. Bismar

**Affiliations:** 1Department of Pathology and Laboratory Medicine, Cumming School of Medicine, University of Calgary, Calgary, AB T2N 1N4, Canada; 2Departments of Oncology, Biochemistry and Molecular Biology, Cumming School of Medicine, Calgary, AB T2N 4N1, Canada; 3Arnie Charbonneau Cancer Institute and Tom Baker Cancer Center, Calgary, AB T2N 4N1, Canada; 4Departments of Mathematical and Statistical Sciences and Medical Oncology, Faculty of Medicine and Dentistry, University of Alberta, Edmonton, AB T6G 2R7, Canada; 5Prostate Cancer Center, Calgary, AB T2V 1P9, Canada

**Keywords:** CDK2AP1, ERG, PTEN, p53, AR, androgen deprivation therapy, cancer-specific mortality, prostate cancer

## Abstract

Prostate cancer (PCa) is one of the most commonly diagnosed types of malignancy and is the second leading cause of cancer-related death in men in developed countries. Cyclin dependent kinase 2 associate protein 1(*CDK2AP1*) is an epigenetic and cell cycle regulator gene which has been downregulated in several malignancies, but its involvement in PCa has not yet been investigated in a clinical setting. We assessed the prognostic value of *CDK2AP1* expression in a cohort of men diagnosed with PCa (n = 275) treated non-surgically by transurethral resection of the prostate (TURP) and studied the relationship between *CDK2AP1* expression to various PCa molecular subtypes (ERG, PTEN, p53 and AR) and evaluated the association with clinical outcome. Further, we used bioinformatic tools to analyze the available TCGA PRAD transcriptomic data to explore the underlying mechanism. Our data confirmed increased expression of *CDK2AP1* with higher Gleason Grade Group (GG) and metastatic PCa (*p* <0.0001). High CDK2AP1 expression was associated with worse overall survival (OS) (HR: 1.62, CI: 1.19–2.21, *p* = 0.002) and cause-specific survival (CSS) (HR: 2.012, CI 1.29–3.13, *p* = 0.002) using univariate analysis. When compared to each sub-molecular type. High CDK2AP1/PTEN-loss, abnormal AR or p53 expression showed even worse association to poorer OS and CCS and remained significant when adjusted for GG. Our data indicates that CDK2AP1 directly binds to p53 using the Co-Immunoprecipitation (Co-IP) technique, which was validated using molecular docking tools. This suggests that these two proteins have a significant association through several binding features and correlates with our observed clinical data. In conclusion, our results indicated that the CDK2AP1 overexpression is associate with worse OS and CSS when combined with certain PCa molecular subtypes; interaction between p53 stands out as the most prominent candidate which directly interacts with CDK2AP1.

## 1. Introduction

Prostate cancer (PCa) is a heterogeneous disease and remains as one of the most common cancers worldwide [1]. As of yet, there are few clinical biomarkers used to estimate the likelihood of lethal PCa and predict disease outcomes. Evaluation of molecular profiles allows a more precise risk classification of PCa [2], and within this classification alteration in *ETS*-*related gene (ERG)* and phosphatase tensin homologue *(PTEN)* are among the most common genomic alterations to occur in PCa [3,4]. *ERG* rearrangements occur in approximately half of PCa cases and may be associated with poor prognosis in some studies [5]. *PTEN* deletions, have been associated with a worse prognosis [6]. Androgen receptor (AR) expression plays a pivotal role in castration-resistant prostate cancer (CRPC) [7]. *TP53* mutation affecting 50% of metastatic PCa cases are well known to serve as a prognostic biomarker of PCa, especially CRPC [8]. However, there remains conflicting results regarding the implementation of biomarker signatures, especially given variance between different ethnicities and the search for an effective biomarker continues [9,10].

Cyclin-dependent Kinase 2-associated Protein 1 (*CDK2AP1*) was originally called *p12^DOC−1^* or *doc-1* (*deleted in oral cancer-1*) is a highly conserved, ubiquitously expressed gene on chromosome 12q24, which encodes a 115-amino acid protein [11,12]. *CDK2AP1* has been reported to function as an epigenetic and cell cycle regulator [13]. Additionally, *CDK2AP1* has been suggested to have a role in TGF-Beta mediated growth suppression and may also be involved in epigenetic gene regulation [14]. Varying levels of *CDK2AP1* have been observed in tumors and adjacent non-cancerous normal tissues. Decreased expression has also been found in cancers such as breast [15], oral squamous cell carcinoma [16] and lung cancer [17]. On the contrary, a few studies reported high *CDK2AP1* mRNA levels associate with tumorigenesis. Xu et al. found that *CDK2AP1* knockdown by RNA interference (RNAi) resulted in reduction in cell proliferation, glioma growth and tumorigenesis. [18]. Furthermore, Gera et al. [15] found 39-fold increased expression of *CDK2AP1* in adjacent non-cancerous normal tissues when compared to breast tumors.

In the present study, we are the first to investigate the role and prognostic value of CDK2AP1 in a clinical setting of large cohort of PCa patients treated non-surgically by transurethral resection of the prostate (TURP) including incidental, advanced and castrate-resistant prostate cancer (CRPC). We evaluated the possible relevance to known molecular subtypes in PCa including ERG, PTEN, AR and p53 and performed early in vitro studies as well as expanded bioinformatic analysis into molecular associations between CDK2AP1 and lethal PCa.

## 2. Materials and Methods

### 2.1. Study Population and Tissue Microarray Construction

The study cohort consisted of a group of men diagnosed with PCa by TURP (n = 275). Patients within this cohort were either not treated actively or treated with Androgen deprivation therapy (ADT) pre or post TURP. Patients with treatments post TURP were categorized as advanced group and those with treatments prior to TURP sample who had advanced local disease with obstructive symptoms while on ADT, were categorized as CRPC group. The incidental group was categorized as patients with no prior ADT therapy and who had Gleason grade group 1–3. The cohort samples were collected between 1999 and 2014, with an overall median follow-up of 39.28 (1.48–101.22) months. Each sample diagnosis and GG was confirmed by two of the study pathologists. Gleason scoring was assessed according to the 2018 WHO and ISUP grade groups. Table 1 demonstrates patient demographics of the study cohort relative to the proposed subgroups, GG and biomarker groups expression. Clinical follow-up was approved by the University of Calgary, Cumming School of Medicine Ethics Review Board from the Alberta Tumor Registry and included dates of therapy, overall survival (OS), and prostate cancer specific survival (CSS). Tissue samples of the cohort were assembled on two tissue microarrays (TMAs) with an average of two cores per patient using a manual tissue arrayer (Beecher Instruments, Silver Spring, MD, USA).

### 2.2. Immunohistochemistry

CDK2AP1 protein expression was assessed using immunohistochemistry (IHC) on a Dako Omnis autostainer. Briefly, about 4 µm formalin-fixed paraffin-embedded sections were pretreated with citrate pH 6.0 epitope retrieval buffer. Mouse monoclonal CDK2AP1 antibody (Cat# SC-390283, Santa Cruz, Dallas, TX, USA) (1:200) and Androgen receptor clone AR27 (1:25) (mouse monoclonal AR (Cat# NCL-L-AR-318, Leica Biosystems Inc. Buffalo Grove, IL 60089, USA). The FLEX DAB+ Substrate Chromogen system was used as the post-incubation detection reagent. PTEN and ERG were evaluated using Fluorescence in situ hybridization (FISH), as previously described [19,20]. We assessed p53 expression using a previously validated method [21].

### 2.3. Pathological Analysis

Histological diagnoses of individual TMA cores were confirmed by the two pathologists. on the initial slides. Gleason scoring was assessed according to the *2018 World Health Organization/International Society of Urological Pathology GGs*. In each patient, the two predominant patterns of PCa were sampled and included on the TMAs for analysis. CDK2AP1 IHC expression was assessed using a four-tiered system (0, negative; 1, weak; 2, moderate; and 3, high expression). PTEN and ERG IHC were evaluated as binary values (negative vs. positive) reflective of ERG gene rearrangements and homozygous PTEN deletions as detected by FISH, as previously described [19,20]. We assessed p53 expression using a previously validated method reflective of *TP53* sequencing mutations [22]. In brief, score 1: wild type; nuclear staining (strong or weak) with internal control; score 0: absent nuclear staining with positive control; score 2: Overexpressed nuclear staining and score 3: Cytoplasmic staining. In short, score 1 = normal and score 0, 2 and 3 = abnormal (but each represents a different type of alteration to p53). AR IHC was assessed using three-tiered system; 0 negative, 1; weak; 2 moderate and 3 high intensities. The AR groups were then combined into low (0–1) and high (2–3) for simplifying analysis.

### 2.4. Cell line and Western Blotting

Cell lines Hek293, Hela, RWPE1, PC3, DU-145, LNCAP and VCAP were obtained from ATCC. The cells were grown and maintained according to ATCC recommended protocols. PC3-ERG stable cell lines received from Felix Feng, University of Michigan and were maintained in our lab in DMEM/F12 with 10% FBS [23].

### 2.5. Protein–Protein Interaction Using Co-Immunoprecipitation and Molecular Docking

For investigating the protein–protein interaction relevance between the PCa molecular subtypes and CDK2AP1, we used Pierce Direct Magnetic IP/CO-IP kit (Cat # 88828, Thermo Scientific, Waltham, MA, USA). Briefly, HEK293 cells were grown in DMEM media supplemented with 10% FBS and 100 U/mL penicillin/streptomycin. Cells were harvested and lysed with IP lyses buffer (Ref#1861603, Thermo Scientific, USA) containing Protease/Phosphatase Inhibitor Cocktail (Cell signaling Cat# 5872S). Pierce NHS-Activated magnetic beads incubated with 5 μg of CDK2AP1 monoclonal mouse antibody (Santa Cruz Cat# sc-390283) or IgG (as a negative control) for 2 h at 4 °C on rotator, followed by incubation with 1500 μg of total protein lysate for 2 h or overnight at 4 °C. Finally, the mixture was washed and eluted. The whole cell lysate (Input) and the immune-precipitant or IgG (negative control) was subjected to Western blotting to confirm the interaction.

To construct protein–protein interaction model, active residues involved in protein structure of p53 (PDB: 2OCJ) [24] and CDK2AP1 (PDB: 2KW6) [25] were utilized. HADDOCK 2.4 developed by the Bonvin lab utilized for protein docking (https://www.bonvinlab.org/software/haddock2.4) accessed on 10 July 2022 [26]. Chain A of p53 and chain B for CDK2AP1 were used to model this interaction. For both molecules, a value of 15% was used as the minimum percentage of relative solvent accessibility (RSA) to consider a residue as accessible. Additionally, 40% was used as the Minimum percentage of RSA to automatically define surface neighbors of active residues as passive. The clustering method used was Fraction of Common Contacts (FCC), with a RMSD cut-off of 0.6 and a minimum cluster size of 4. PyMol software used to model the protein structure and interactions. For the physical contacts, a distance of 4 angstroms were used as the cut-off for classifying an interaction as physical interaction. Furthermore, for modeling the interaction we used the following p53 active residues, amino acids specified in the two transactivation’s domains, TAD (residues 6–30) and TAD2 (residues 35–59), p53 domain (residues 99 to 289) and tetramerization domain (residues 319–358).

### 2.6. CDK2AP1 mRNA Expression in TCGA PRAD and Associated Gene Set Enrichment Analysis

The gene signature analysis was obtained using TCGA PRAD transcriptomics data [27]. Pan-cancer analysis used to analyze the expression range of *CDK2AP1* gene in tumor vs. normal across TCGA, GTex and TARGET databases and includes 22 tissue types simultaneously. This tool based on RNA-seq- rapid analysis servers to furnish comparative information for a selected gene of interest. We further compared the tumor gene array data paired with either normal noncancerous or adjacent normal tissue. Then, the results were blotted and presented with the proportions of tumor samples that show higher expression of the selected gene compared to normal samples at each of the quantile cut-off values (minimum, 1st quartile, median, 3rd quartile, maximum) [27]. Then, we used TCGA PRAD mutations profiling of *PTEN, TP53* and *AR* to assess *CDK2AP1* expression and compared *CDK2AP1* between mutated and un-mutated samples.

To further elaborate the *CDK2AP1* gene potential mechanism in PCa, we used linkedOmics tools (http://www.linkedomics.org (accessed on 10 July 2022)) to analyze the cohort available in TCGA PRAD [28]. Both *CDK2AP1* overrepresented (ORA) and gene set enrichment (GSEA) were analyzed based on FDR and categorized as molecular function, cellular components and biological process using WEB-based GEne SeT AnaLysis Toolkit and Explorer [29].

### 2.7. Statistical Analysis

SPSS used to conduct clinical data statistical analysis (IBM SPSS Statistics for Windows, version 25.0, released 2017; IBM Corp., Armonk, NY, USA). Frequency and proportions were reported for categorical data. Mean and standard deviations were reported for normally distributed continuous data, and median and range were reported for non-normally continuous data. Chi-square test was used to compare two categorical variables and Fisher’s exact test was used where the cell frequencies were <5. Overall survival (OS) was calculated from the time of diagnosis to death; patients who were alive at the end of the study period were censored. Prostate cancer–specific survival (CSS) was defined as death due to PCa; patients who died due to any other reason or who were alive at the end of the study period were censored. OS and CSS were analyzed using the Kaplan–Meier method. Median time and the corresponding 95% confidence intervals (CI) were reported. Log-rank tests were used to compare two or more survival curves. Univariate Cox’s proportional hazard models were used to determine the factors associated with OS and CSS; hazard ratios (HR) and the corresponding 95% CIs were reported. Adjusted Cox’s models for Gleason score variable were reported as well, in addition to the univariate association. A *p* value of <0.05 was used for statistical significance, and two-sided tests were utilized.

## 3. Results

### 3.1. CDK2AP1 Expression in PCa

In our cohort, high CDK2AP1 expression was noted in 141/275 (51.3%) of cases and 79.4% of these cases were either advanced or castrate resistant. CDK2AP1 IHC intensity had mean expression of 0.71 ± 0.93 in incidental, 1.49 ± 1.17 in advanced and, 1.63 ± 1.06 in castrate resistant group (*p* < 0.0001) (Figure 1A,B).

### 3.2. High CDK2AP1 Expression Is Associated with Higher Gleason Group (GG) and Metastatic Disease

Separately, high CDK2AP1 expression, *PTEN*-loss, ERG negativity, and wild type p53 expression were associated with higher Gleason Groups in this cohort (*p* < 0.0001, *p* < 0.0001, *p* < 0.0001, *p* < 0.001, respectively). Furthermore, a significant association was observed between higher Gleason Groups when combining CDK2AP1 expression with other PCa biomarkers, (high CDK2AP1 expression/PTEN loss, high CDK2AP1/AR expression, high CDK2AP1/ERG positive, and high CDK2AP1/abnormal p53 expression (*p* < 0.0001, *p* < 0.0001, *p* < 0.0001, *p* < 0.0001, respectively) (Table 1).

### 3.3. High CDK2AP1 Expression Is Associated with Metastatic Prostate Cancer

Separately in this cohort, high CDK2AP1 and high AR expression were significantly associated with metastatic disease (*p* value 0.034 and 0.019, respectively) and when combined, there is a higher rate of metastatic disease observed in high CDK2AP1 and high AR expression cases (*p* value 0.013). However, this relationship was not significant for CDK2AP1 when combined with other PCa subtypes. Notably, the mean intensity of CDK2AP1 was significantly higher for metastatic (1.56 ± 1.07) versus non-metastatic (1.19 ± 1.14) disease (*p* value 0.033). Appendix A.

### 3.4. High CDK2AP1 Expression Is Associated with Poor Overall and Cause Specific Survival in PCa

High CDK2AP1 expression was significantly associated with poor OS (HR: 1.62, CI: 1.19–2.21, *p* = 0.002) and poor CSS (HR: 2.01, CI: 1.29–3.13, *p* = 0.002) (Figure 1C,D). Moreover, high CDK2AP1 expression was noted in 24% of PTEN negative cases, 18.2% of ERG positive cases, 3.6% AR low expression cases and in 14.2% of abnormal p53 cases (Table 1).

### 3.5. High CDK2AP1 Expression in Combination with ERG, PTEN, p53 or AR Signify Poorer Overall and Cause Specific Survival

When high CDK2AP1 expression was combined with other PCa molecular subtypes (Figure 2), *PTEN*-loss and high CDK2AP1 expression were even more significantly associated with poorer OS (HR: 2.83, CI: 1.89–4.24, *p* < 0.0001) and CSS (HR: 4.43, CI: 2.28–7.93, *p* < 0.0001) compared to each marker individually or any other combination of the two specifically in relation to CCS in univariate analysis (Table 2). However, *PTEN*-loss and high CDK2AP1 expression had a higher CSS HR compared to *PTEN*-loss and low CDK2AP1 expression cases (HR: 4.18, CI: 2.07–8.42, *p* < 0.0001). In addition, our data further indicate that combining CDK2AP1 expression with other biomarkers such as AR, ERG or p53 status is associated with added prognostic value compared to individual markers.

Interestingly, abnormal p53 combined with high CDK2AP1 expression was the most significant combination, significantly associated with poorer OS (HR: 3.81, CI: 2.46–5.90, *p* < 0.0001) and CSS (HR: 6.01, CI: 3.36–10.76, *p* < 0.0001). More importantly, this relationship was still significant in multivariate analysis, after adjusting for Gleason grade grouping; OS (HR: 2.29, CI: 1.42–3.70, *p* = 0.001) and CSS (HR: 2.36, CI: 1.28–4.34, *p* = 0.006). However, other combined biomarkers signatures showed comparable prognostic values, depending on outcome measured (Table 3).

### 3.6. CDK2AP1 Interacts with p53

Co-Immunoprecipitation (Co-IP) was used to explore possible direct interactions between CDK2AP1 and different PCa molecular subtypes. Initially, we screened several PCa cell line (Hela and Hek293) to check whether they express CDK2AP1 and the other molecular subtypes. We found that Hek293 and Hela cells expressed both CDK2AP1 and p53 (Figure 3A). Co-IP results revealed that p53 potentially immunoprecipitated with CDK2AP1 but PTEN, ERG or AR did not (Figure 3B). Further, we used protein–protein molecular docking tools to validate the potential binding sites between CDK2AP1 and p53.

When analyzing the docking results obtained from the HADDOCK 2.4 webs server, the lowest score was selected as the best model (55.8 ± 18.0, this cluster was also shown to have a z-score of −1.5) which interpreted the highest possible interaction.

CDK2AP1 protein structure is comprised of 115 amino acid residues, which come together to form a structure with two alpha helices. Each monomer of CDK2AP1 undergoes dimerization to form an active protein. This protein only contains one active domain which lies between residue 3 and 114, this is hypothesized as being the active site domain that interacts with a variety of other proteins. The p53 monomer structure utilized is modelled in the absence of DNA consists of 2 alpha helices and 8 beta-strands, the monomers undergo tetramerization to form an active p53 protein. The active residues of this p53 protein include two transactivation’s domains (TADs), the first TAD is between residues 6–30, while the second TAD is between residues 35–59. Moreover, the p53 active site domain is situated between residues 99 and 28. The protein also contains a tetramerization domain which is found between residues 319–358.

Visualizing the modelled interaction (Figure 3C), we were able to detect the presence of hydrogen bonding and physical contact interactions between CDK2AP1 and p53. From the in silico derivation, it was observed that CDK2AP1 exhibited 15 hydrogen bonding contacts with p53 residues. Additionally, the modelled interactions also demonstrated 29 physical contacts within 4 angstroms of interacting p53 residues (Appendix A). In the visualization of the docking, we found hydrogen bonding to be present between the N-terminal histidine residues (H3, H8, H10) with side chains of glutamate, glutamine, serine, and histidine residues found on p53 (Appendix A). The remainder of the hydrogen bonding interactions were between the residues found on the alpha-helixes of CDK2AP1 (between S11–G44) with various residue sidechains spread throughout the p53 molecule. Next, a significant number of physical contacts were found to be present when analyzing this protein interaction model, the exact interactions have been listed in Appendix A. The M1 and K25 residues of CDK2AP1 were shown to have physical contacts with 6 of the p53 residues. Furthermore, two arginine residues (R40 and R43) on the second alpha-helix motif were shown to have significant physical interaction with p53 residues. Additionally, moderate amounts of physical contact were also observed between histidine residues 3, 8, and 10 with each residue interacting with 4–5 other residues of p53.

Moreover, this modelled protein–protein interaction had a *Van der Waals* energy value that was calculated to be −72.1 ± 14.0, along with electrostatics energy value of −521.8 ± 6.8. In addition to this, the modelled interaction also had a desolvation and restraints violation energy value of −15.4 ± 0.8 and 2476.1 ± 34.4, respectively.

### 3.7. CDK2AP1 mRNA Expression in Pan-Cancer Data

Pan-Cancer data analysis for *CDK2AP1* mRNA expression in 22 tumors revealed *CDK2AP1* was significantly upregulated in all 22 tumors including PCa (Figure 4A). Interestingly, the significant difference was more apparent when the tumor was compared to noncancerous normal samples vs. adjacent normal tissue in prostate samples (Figure 4B,C). PRAD TCGA mutation profiling for the *PTEN*, *TP53* and *AR*, we found *CDK2AP1* is highly expressed in *PTEN* and *TP53* but not *AR* mutations (Figure 4D).

### 3.8. CDK2AP1 Gene Set Enrichment Analysis in PCa TCGA PRAD

*CDK2AP1* Gene Set Enrichment analysis data revealed a significant and distinctive gene set profiling as presented by volcano blots (Figure 5A). Heat map (Figure 5B) showing the top 50 positively and top 50 negatively correlated genes (Figure 5C). *CDK2AP1* overexpression GSEA the genes set enrichment analysis (Figure 5D) indicated upregulating biological processes including cell proliferation and growth, upregulation in metabolic process, biological regulations, cellular components organization, localization, development process, cell communication. The Cellular component upregulated genes appear mostly in the nucleus (1686 genes) and then followed by the cytoplasmic membrane (1593 genes). The positive correlated genes for molecular functions show a wide range of potential functions from protein and nucleic acid binding to chromatin binding. When we analyzed the overrepresented (ORA) genes (Figure 5E), many cellular functions such as mitochondria respiratory chain complex assembly, mitochondria gene expression, mRNA processing, ribonucleoprotein complex biogenesis, protein localization to endoplasmic reticulum, DNA-template transcription, and termination, tRNA metabolic process, protein-DNA complex subunit organization, RNA localization and protein localization to mitochondria were overrepresented. Interestingly, the negative correlated gene sets were mostly involved declined immune and inflammatory responses which were downregulated. All together, these data indicated a potential function of *CDK2AP1* in tumorigenesis and immune-evading mechanism.

## 4. Discussion

*CDK2AP1* has diverse functions ranging from epigenetics moderator to cell cycle regulator [13]. Earlier studies suggest that *CKD2AP1* may function as a tumor suppressor, however, we saw a unique *CDK2AP1* overexpression pattern in prostatic adenocarcinoma along with most other malignancies. Our data indicates poor survival associated with increasing expression of CDK2AP1 in patients with PCa. Interestingly, this association was variable when implicated with other molecular subtypes of PCa (i.e.,: when combined with PTEN, ERG, AR and p53). Previous studies have shown the association between *PTEN*-loss and high Gleason scores [30], however the role of CDK2AP1 with PTEN and other various PCa subtypes remains a mystery. In this cohort, high CDK2AP1 expression with *PTEN*-loss, high AR, positive ERG and abnormal p53 expression were significantly associated with higher Gleason Groups and poor overall survival. Furthermore, there was a significant increase in the mean intensity of CDK2AP1 expression in metastatic disease when compared to non-metastatic disease. Overall, the higher expression of CDK2AP1 in high Gleason Group and metastatic tumors demonstrate its role as a potential biomarker for lethal PCa. Our data suggested the role of CDK2AP1 as a possible oncogene in this malignancy.

Even though PCa is still trailing other cancers in biomarker guided treatment, genes such as *PTEN, ERG, AR* and *TP53* have been noted as clinically relevant biomarkers to predict patient’s outcomes and suggest therapeutic strategies [4,31]. p53 inactivation or abnormal expression has been demonstrated to be associated with adverse patient clinical outcomes and metastatic disease while high expression of ERG and PTEN-loss have been demonstrated to be clinically relevant to shorter patient survival, higher Gleason scores and unfavorable pathological and clinical outcomes [32]. We found an interesting association between CDK2AP1 and p53 expression in our clinical data which were confirmed by Co-IP and protein–protein interaction docking tools. Alsayegh et al. demonstrated *CDK2AP1* knockdown downregulated p53 in primary human fibroblasts [33]. Although p53 is classically thought of as a tumor suppressor, recent studies have demonstrated that some types of mutant TP53 display gain-of-function with increased expression of malignant properties [34].

We saw a significant overexpression of *CDK2AP1* in PCa tumors through mRNA-seq data when compared to normal tissue, however, this significance was not persistent when compared to adjacent normal tissues (Figure 4C). More interestingly, this significance was observed with *PTEN* and *TP53* mutated samples. In breast cancer, Gera et al. [15] reports that they found 39-fold increase of normalized mRNA expression of *CDK2AP1* in noncancerous adjacent tissues [15]. Moreover, Xu et al. [18] studied *CDK2AP1* overexpression association with tumorigenesis in glioma in vitro and in vivo and found that, *CDK2AP1* knockdown inhibits cells proliferation, and colony formation. Furthermore, *CDK2AP1* knockdown induced cell cycle arrest and apoptosis. Furthermore, an increased *CDK2AP1* expression was reported in precursor lesions of invasive tumors in the oral epithelium [35]. This data supports with our findings in PCa.

Many cellular functions were found to be significantly over and under-represented. Mostly, pathways directly related to DNA-templated transcription, termination, mRNA processing, RNA localization, and mitochondrial respiratory chain complex assembly and mitochondrial gene expression were over-represented. Such pathways correspond to the oncogenic role of *CDK2AP1* in cellular proliferation as mitochondrial dysfunction and dysregulation is very common in PCa behavior [36]. Furthermore, pathways directly related to the immune response against tumor cells such as leukocyte activation involved in inflammatory response, regulation of leukocyte activation, interleukin-6 production were under-represented in *CDK2AP1* gene set enrichment analysis. Previous studies have shown that leukocyte-to-monocyte ratio is higher in patients with prostate cancer which correspond to the under-represented regulation of leukocyte activation [37]. Furthermore, it has been described that interleukin-6 is down regulated in human prostatic carcinoma cells which provides further evidence for this enrichment gene analysis and role of CDK2AP1 as an oncogene. It appears that *CDK2AP1* oncogenic overexpression may enhance the ability of tumor cells to evade the immune system [38].

Some limitations of this study, relate to the cohort investigated being a heterogenous population ranging from incidental PCa to advanced and CRPC in addition to samples being TURP in nature. Nonetheless, this represents a real sample of patients at a tertiary institution in Western country. Further in vitro studies are needed to elucidate more of CDK2AP1 function in tumorigenesis.

## 5. Conclusions

In conclusion, our study confirms a significant role of CDK2AP1 in PCa. CDK2AP1 upregulation was shown to be associated with poor OS and CSS in PCa. Furthermore, when combined with other PCa molecular subtypes, p53 and PTEN were the most prevalent mutations associated with CDK2AP1.

## Figures and Tables

**Figure 1 ijms-23-13326-f001:**
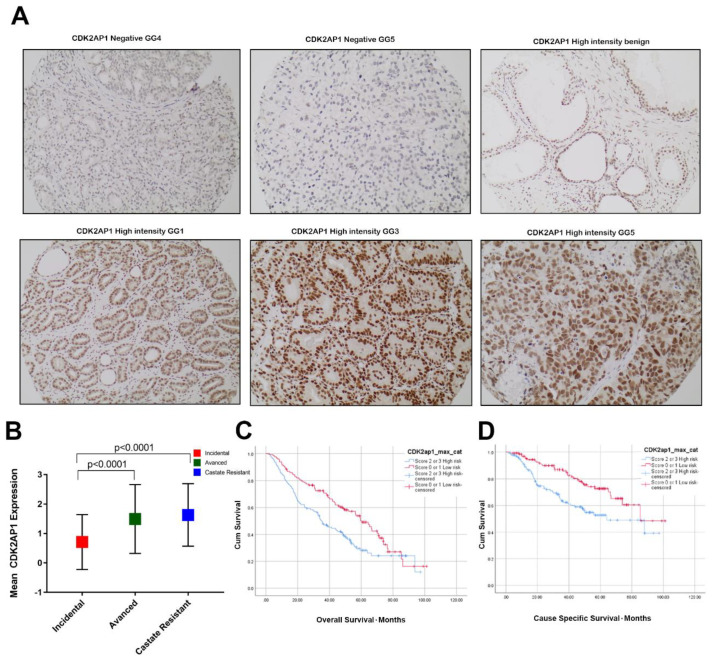
The expression of CDK2AP1 in Prostate tissues and associated survival analysis (**A**) IHC staining of CDK2AP1 in benign and PCa tissues (20× magnification). (**B**) Box plots represents the mean expression of CDK2AP1 in Incidental, Advanced, and CRPC in our study cohort samples. The error bars indicate the standard deviation of the mean (SD). Student *t*-test was performed, *p* value < 0.05 was considered significant between Advanced or CRPC compared with Incidental (Benign). (**C**) Kaplan–Meier (KM) curves outlining the expression of CDK2AP1 in relation to overall survival (OS) and (**D**) Cause-specific survival (CSS).

**Figure 2 ijms-23-13326-f002:**
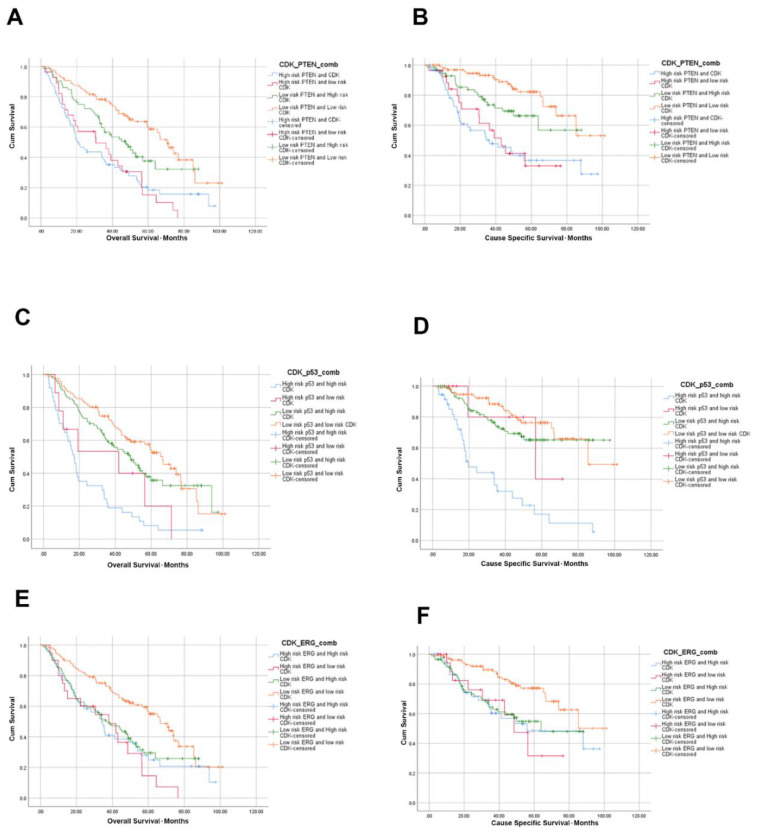
KM curves demonstrating the relationship between CDK2AP1 expression and OS and CSS (**A**,**B**) CDK2AP1 and PTEN expression. (**C**,**D**) CDK2AP1 and p53 expression. (**E**,**F**) CDK2AP1 and ERG expression. Samples were scores semi-quantitatively using a four-tiered system for CDK2AP1 (negative—0; weak—1; moderate—2, strong—3; or in short low risk = 0, 1 and high risk = 2, 3) and p53 (wildtype—1; absent nuclear staining—0; overexpressed nuclear staining—2; cytoplasmic staining—3; or in short, low risk = 1 and normal, high risk = 0, 2, 3 and abnormal). PTEN low risk = PTEN positive staining; PTEN high risk = PTEN loss and negative staining. ERG low risk = ERG negative staining; ERG high risk = ERG positive staining. Number of patients at risk given in Appendix A.

**Figure 3 ijms-23-13326-f003:**
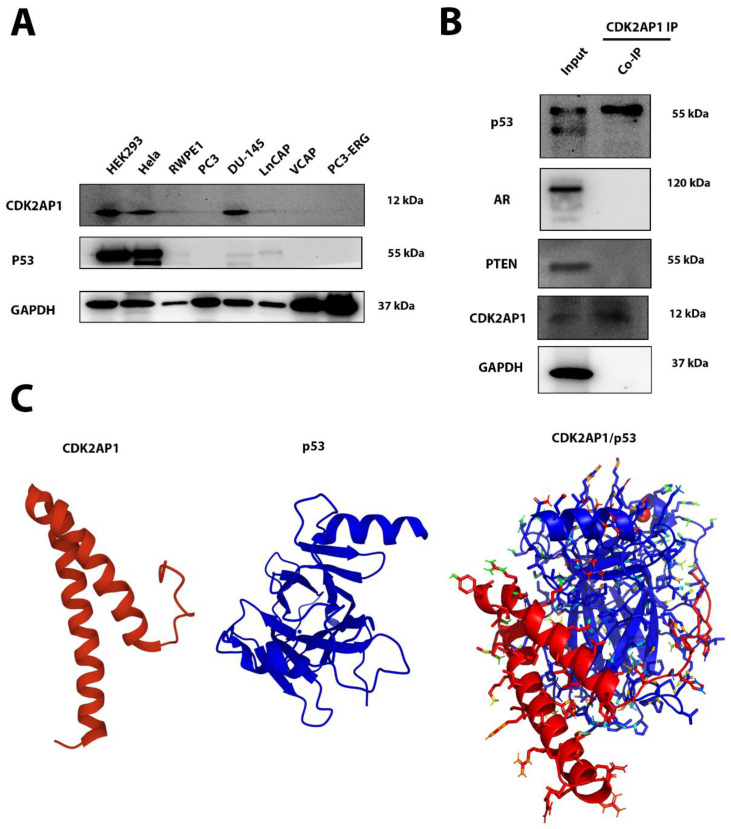
CDK2AP1 potential binding target for p53 (**A**) Western blot screening of CDK2AP1 and p53 protein expression. GAPDH used as loading control. (**B**) The CO-Immunoprecipitation analysis of CDK2AP1 with PCa molecular subtypes (p53, AR, PTEN and ERG). (**C**) Molecular docking module shows CDK2AP1(RED) and p53 (Blue) Protein–protein interaction.

**Figure 4 ijms-23-13326-f004:**
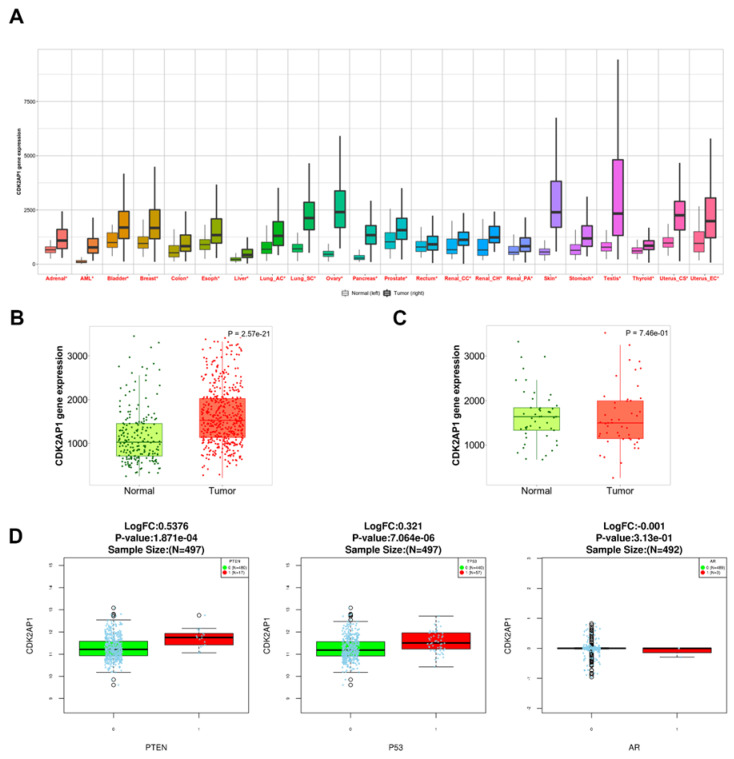
Differential expression of *CDK2AP1* in tumors and normal tissues (**A**) Boxplots shows the *CDK2AP1* genes expression in most of common tumor types Normal (left) Tumor(right). Significant differences by a *Mann–Whitney U test* are marked with red color (* *p* < 0.01). (**B**) Boxplot shows *CDK2AP1* RNA-seq expression in PCa tumor vs. noncancerous normal tissue. (**C**) and PCa tumor vs. Paired adjacent-normal tissue and (**D**) *CDK2AP1* expression in *PTEN/TP53* or *AR* mutations (green is wildtype and red is mutated).

**Figure 5 ijms-23-13326-f005:**
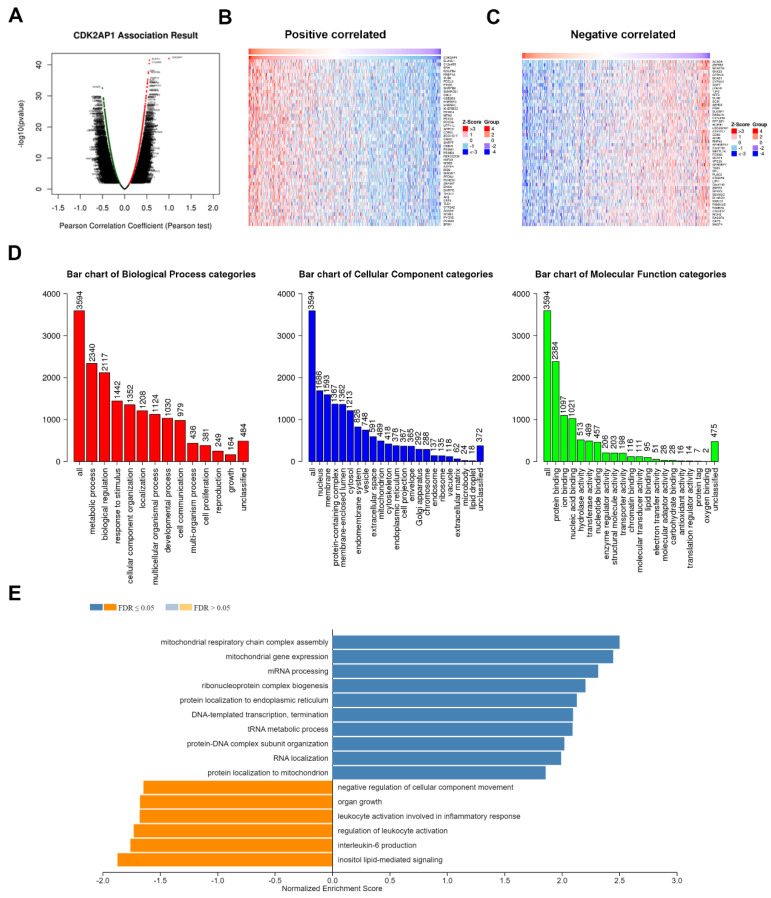
*CDK2AP1* gene set enrichment association in PCa tumorigenesis (**A**) Volcano blots showed the *CDK2AP1* associated genes in TCGA PRAD. Significant up-regulation genes (dark red dots, *FDR* < 0.01) whereas genes showed significant down-regulation (dark green dots, FDR < 0.01). (**B**,**C**) Heatmap represents the top 50 positively or negatively correlated genes associated with *CDK2AP1* overexpression. (**D**) Bar blots represents GSEA categories; Biological Process, Cellular Component and Molecular Function (red, blue and green bars, respectively). The height of the bar represents the number of IDs in the user list and in the category. The FDR is calculated using *Benjamini-Hochberg* method. (**E**) Bars blot showed the positively correlated GO (Molecular Function) for Overrepresentation Enrichment Analysis (ORA). Data were analyzed using Pearson correlation coefficient and ranked according to FDR value.

**Table 1 ijms-23-13326-t001:** Demographics of TURB cohort: clinical and pathological biomarkers.

Variables	Total (n = 275) (%)
Gleason Score	
Grade group 1	82 (29.8)
Grade group 2 (3 + 4)	29 (10.5)
Grade group 3 (4 + 3)	17 (6.2)
Grade group 4 (8)	20 (7.3)
Grade group 5 (9/10)	121 (44.0)
Missing	6 (2.2)
Deceased	
Yes	165 (60.0)
No	108 (39.3)
Missing	2 (0.7)
Prostate cancer specific mortality	
Yes	85 (30.9)
No	188 (68.4)
Missing	2 (0.7)
Cancer Subgroup	
Incidental	94 (34.2)
Advanced	109 (39.6)
Castrate Resistant	72 (26.2)
CDK2AP1 Score (Score by Cancer subgroup)	
Score 2 or 3	141 (51.3%)
Incidental Score	29 (10.5)
Advanced Score	66 (24.0)
Castrate Resistant Score	46 (16.7)
Score 0 or 1	134 (48.7)
Incidental Score	65 (23.6)
Advanced Score	43 (15.6)
Castrate Resistant Score	26 (9.5)
PTEN and CDK2AP1 combined	
PTEN negative and CDK2AP1 Score 2, 3	66 (24.0)
PTEN negative and CDK2AP1 Score 0, 1	28 (10.2)
PTEN positive and CDK2AP1 score 2, 3	73 (26.5)
PTEN positive and CDK2AP1 Score 0, 1	102 (37.1)
Missing	6 (2.2)
ERG and CDK2AP1 Combined	
ERG positive and CDK2AP1 Score 2, 3	50 (18.2)
ERG positive and CDK2AP1 Score 0, 1	20 (7.3)
ERG negative and CDK2AP1 Score 2, 3	89 (32.4)
ERG negative and CDK2AP1 Score 0, 1	110 (40.0)
Missing	6 (2.2)
AR and CDK2AP1 Combined	
AR Score 0, 1 and CDK2AP1 Score 2, 3	10 (3.6)
AR Score 0, 1 and CDK2AP1 Score 0, 1	12 (4.4)
AR Score 2, 3 and CDK2AP1 Score 2, 3	126 (45.8)
AR Score 2, 3 and CDK2AP1 Score 0, 1	115 (41.8)
Missing	12 (4.4)
p53 and CDK2AP1 Combined	
p53 Score 0, 2, 3 and CDK2AP1 Score 2, 3	39 (14.2)
p53 Score 0, 2, 3 and CDK2AP1 Score 0, 1	10 (3.6)
p53 Score 1 and CDK2AP1 Score 2, 3	98 (35.6)
p53 Score 1 and CDK2AP1 Score 0, 1	99 (36.0)
Missing	29 (10.5)

CDK2AP1, AR scores as 0; negative, 1; weak, 2; moderate and 3, high intensity. p53 scores 0; absent nuclear staining, 1 wild type, 2; overexpressed nuclear staining, 3; cytoplasmic staining; PTEN scores 0; negative, 1; positive.

**Table 2 ijms-23-13326-t002:** Univariate analysis of CDK2AP1 expression in relation to OS and CSS.

Variables	Overall Survival		Prostate Cancer Specific Survival	
	HR (95% CI)	*p*-Value	HR (95% CI)	*p*-Value
CDK2AP1 (Score 0, 1) (n = 129)				
Score 2, 3 (n = 135)	1.62 (1.19–2.21)	0.002	2.01 (1.29–3.13)	0.002
PTEN (Positive) (n = 170)				
PTEN negative (n = 89)	2.44 (1.79–3.33)	<0.0001	3.26 (2.12–5.01)	<0.0001
CDK2AP1 and PTEN (PTEN positive and CDK Score 0, 1) (n = 98)				
PTEN negative and CDK score 2, 3 (n = 62)	2.83 (1.89–4.24)	<0.0001	4.43 (2.28–7.93)	<0.0001
PTEN negative and CDK score 0, 1 (n = 28)	3.07 (1.87–5.01)	<0.0001	4.18 (2.07–8.42)	<0.0001
PTEN positive and CDK score 2, 3 (n = 72)	1.64 (1.08–2.51)	0.021	1.97 (1.05–3.71)	<0.0001
CDK2AP1 and ERG (ERG negative and CDK score 0, 1) (n = 106)				
ERG positive and CDK score 2, 3 (n = 48)	2.00 (1.31–3.06)	0.001	2.42 (1.33–4.40)	0.004
ERG positive and CDK score 0, 1 (n = 20)	2.62 (1.51–4.53)	0.001	2.63 (1.18–5.88)	0.018
ERG negative and CDK score 2, 3 (n = 86)	1.85 (1.27–2.69)	0.001	2.28 (1.34–3.87)	0.002
CDK2AP1 and AR (AR score 2, 3 and CDK score 0, 1) (n = 110)				
AR score 0, 1 and CDK score 2, 3 (n = 10)	3.53 (1.73–7.17)	0.001	5.41 (2.19–13.37)	<0.0001
AR score 0, 1 and CDK score 0, 1 (n = 12)	2.09 (1.06–4.12)	0.033	2.81 (1.14–6.92)	0.025
AR score 2, 3 and CDK score 2, 3 (n = 120)	1.74 (1.24–2.44)	0.001	2.20 (1.34–3.62)	0.002
CDK2AP1 and p53 (p53 score 1 and CDK score 0, 1) (n = 96)				
p53 score 0, 2, 3 and CDK score 2, 3 (n = 37)	3.81 (2.46–5.90)	<0.0001	6.01 (3.36–10.76)	<0.0001
p53 score 0, 2, 3 and CDK score 0, 1 (n = 9)	2.37 (1.08–5.24)	0.033	1.49 (0.35–6.35)	0.589
p53 score 1 and CDK score 2, 3 (n = 94)	1.34 (0.92–1.97)	0.130	1.44 (0.82–2.54)	0.203

CDK2AP1, AR scores as 0; negative, 1; weak, 2; moderate and 3; high intensity. p53 scores 0; absent nuclear staining, 1; wild type, 2; overexpressed nuclear staining, 3; cytoplasmic staining.

**Table 3 ijms-23-13326-t003:** Multivariate analysis of CDK2AP1 expression in relation to OS and CSS adjusted for Gleason Score.

Variables	Overall Survival		Prostate Cancer Specific Mortality	
	HR (95% CI)	*p*-Value	HR (95% CI)	*p*-Value
CDK2AP1 and PTEN (PTEN positive and CDK2AP1 Score 0, 1) (n = 91)				
PTEN negative and CDK2AP1 score 2, 3 (n = 60)	1.74 (1.12–2.71)	0.014	1.84 (0.99–3.42)	0.054
PTEN negative and CDK2AP1 score 0, 1 (n = 26)	2.36 (1.41–3.93)	0.001	2.39 (1.14–5.00)	0.021
PTEN positive and CDK2AP1 score 2, 3 (n = 68)	1.44 (0.93 –2.22)	0.102	1.54 (0.80–2.94)	0.193
ERG and CDK2AP1 (ERG negative and CDK2AP1 score 0, 1) (n = 98)				
ERG positive and CDK2AP1 score 2, 3 (n = 46)	1.22 (0.77–1.93)	0.390	1.06 (0.56–2.01)	0.861
ERG positive and CDK2AP1 score 0, 1 (n = 19)	2.30 (1.32–4.01)	0.003	2.08 (0.92–4.71)	0.078
ERG negative and CDK2AP1 score 2, 3 (n = 82)	1.58 (1.07–2.33)	0.021	1.79 (1.04–3.10)	0.037
AR and CDK2AP1 (AR score 2, 3 and CDK2AP1 score 0, 1) (n = 104)				
AR score 0, 1 and CDK2AP1 score 2, 3 (n = 10)	2.07 (1.00–4.29)	0.050	2.17 (0.87–5.42)	0.098
AR score 0, 1 and CDK2AP1 score 0, 1 (n = 11)	1.40 (0.68–2.87)	0.361	1.27 (0.48–3.40)	0.629
AR score 2, 3 and CDK2AP1 score 2, 3 (n = 115)	1.32 (0.92–1.89)	0.128	1.32 (0.79–2.22)	0.294
P53 and CDK2AP1 (p53 score 1 and CDK2AP1 score 0, 1) (n = 91)				
p53 score 0, 2, 3 and CDK2AP1 score 2, 3 (n = 37)	2.29 (1.42–3.70)	0.001	2.36 (1.28–4.34)	0.006
p53 score 0, 2, 3 and CDK2AP1 score 0, 1 (n = 9)	1.51 (0.67–3.40)	0.325	0.63 (0.15–2.72)	0.631
p53 score 1 and CDK2AP1 score 2, 3 (n = 88)	1.07 (0.71–1.60)	0.757	0.89 (0.49–1.61)	0.697

CDK2AP1, AR scores as 0; negative, 1; weak, 2; moderate and 3; high intensity. p53 scores 0; absent nuclear staining, 1; wild type, 2; overexpressed nuclear staining, 3; cytoplasmic staining.

## Data Availability

Not applicable.

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
