# Peer review of "The Association between Cyclin Dependent Kinase 2 Associated Protein 1 (CDK2AP1) and Molecular Subtypes of Lethal Prostate Cancer"

_ijms, 2022, doi:10.3390/ijms232113326_

Round 1

Reviewer 1 Report

The case definitions are not standard

"The study cohort consisted of a group of men diagnosed with PCa by TURP (n = 275). Patients within this cohort were either not treated actively or treated with ADT pre or post TURP. Patients with treatments post TURP were categorized as advanced group and those with treatments prior to TURP sample who had advanced local disease with obstructive symptoms while on ADT, were categorized as CRPC group. The incidental group was categorized as  patients with no prior ADT therapy and who had Gleason grade group 1-3."

Is it not possible for study authors to revisit clinical records and classify patients based on standard definitions based on clinical and imaging criteria. This lacuna is potentially invalidating all clinical conclusions

Please mention number of patients in each group in tables 2 and 3. Low number of patients could point towards potential inflated observed hazard ratios

 what was the agreement between pathologists on scoring the gene expressions. inter rater and test retest agreements must be assessed to say in a valid manner that such a scoring system is reproducible and applicable.

please give the number of patients at risk table below each survival graph where possible.

The bioinformatics aspect of paper is all right

Author Response

Reviewer 1:

The case definitions are not standard

"The study cohort consisted of a group of men diagnosed with PCa by TURP (n = 275). Patients within this cohort were either not treated actively or treated with ADT pre or post TURP. Patients with treatments post TURP were categorized as advanced group and those with treatments prior to TURP sample who had advanced local disease with obstructive symptoms while on ADT, were categorized as CRPC group. The incidental group was categorized as patients with no prior ADT therapy and who had Gleason grade group 1-3."

Is it not possible for study authors to revisit clinical records and classify patients based on standard definitions based on clinical and imaging criteria. This lacuna is potentially invalidating all clinical conclusions

Response: We thank the reviewer for his comment. As the nature of the cohort is retrospective and spans over significant period with earlier clinical management criteria, it is not possible to find additional imaging done for most patients during that period. Additionally, the definition we used is based on defined clinical and pathological criteria, for incidental, advanced and castrate resistant disease or those who had ADT performed for obstructive symptoms. Additionally, since our biomarkers association was significant in the overall cohort, this would not impact the interpretation of data in our opinion, as we did not observe changes in biomarkers association between subgroups.

Please mention number of patients in each group in tables 2 and 3. Low number of patients could point towards potential inflated observed hazard ratios

Response: Thanks for pointing that out, we have updated tables 2 and 3 in the paper. We have added the number for each of the categories. We added thesamples size for the groups and although as anticipated by the reviewer, some sample groups numbers are small resulting in inflated 95% confidence intervals for the hazard ratio, the two main groups as highest risk and lower risk remains of considerable size, supporting significant HR..

What was the agreement between pathologists on scoring the gene expressions. inter rater and test retest agreements must be assessed to say in a valid manner that such a scoring system is reproducible and applicable.

Response: The IHC evaluation is standard practice and only rarely disagreement would be observed and it would be whether a case is called moderate or high (2 vs 3) or negative vs weak (0 vs 1). However, since our findings combine those two readouts in one group, this would not have any effect on resenting our data as we presented the data as binary (neg/weak vs mod/high).

please give the number of patients at risk table below each survival graph where possible.

The bioinformatics aspect of paper is all right

Response: The addition of the number of patients at risk will make the survival graphs even more busy as each of the graphs has four categories, hence, we had previously decided not to include the number at risk in the figure.

We have added the number at risk as a separate table and if you think it to be appropriate then we added this table as a supplementary table and referenced it in the paper.

Reviewer 2 Report

Reviewer Comments and Recommendations

This study has good merit. The objectives of the study are explicit and valid. The Cyclin Dependent Kinase 2 Associated Protein 1 (CDK2AP1) is a ubiquitous protein and its roles are not without controversy in the literature over time; this presents a challenge to the investigator. Here, the authors have shown that there is a statistically significant overexpression of CDK2AP1 associated with overall survival and cause-specific survival for specific molecular subtypes of prostate cancer (Pca) in the clinical setting. Of particular interest is the evidence for a direct p53-CDK2AP1 interaction (further studies will be needed to delineate the mechanism and consequences of an interaction).

I believe the diversity of PCa patient population is off set by the large cohort (n = 275); the findings will be interest to the pathology community, oncologists, and the wider community of those with research interests in cancer biomarker validation, beyond PCa.

The Materials and Methods are largely standard, relevant to the study objectives, appropriate and described clearly and adequately in detail.

The results of the study are for the most part presented well, and the interpretations are supported by evidence-based data, without overinterpretation. The statistical treatment of the data is sound.

There is an issue beginning on page 10, line 286. This refers the reader to Supplementary tables 2 and 3. However, Supplementary Table 2 (contents not re docking study) page 16 of the manuscript is an exact repeat of Table 1. Obviously, this requires correction. Furthermore, there is no Supplementary Table 3 in the manuscript. Consequently, the supporting information relating to the molecular modelling is not presented. No doubt the data exists but given the importance of the finding on the p53- CDK2AP1 interaction, this should be rectified as a priority.

In addition, it would be helpful to state the number of iterations that were applied in the in-silico derivation of the energy-minimised modelled structure of the p53 – CDK2AP1 interaction.

It is recommended that the docking diagram Figure 3 (C) is displayed physically larger in the manuscript; this would be normal practice and will allow for identification of key amino acid residues (indicated by arrows) that are regarded as the key, strongest interactions.

General comment. Although abbreviations are defined immediately after they appear in the text of the manuscript, it is recommended that a separate abbreviation list is included (subject to editorial policy); this should be helpful to those non-PCa specialists unfamiliar with some terms, either new to the area or are part of the wider biomarker research community to whom this paper will be of interest.

My recommendation is that the article merits publication subject to amendments outlined above.

Author Response

Reviewer2:

Reviewer Comments and Recommendations

This study has good merit. The objectives of the study are explicit and valid. The Cyclin Dependent Kinase 2 Associated Protein 1 (CDK2AP1) is a ubiquitous protein, and its roles are not without controversy in the literature over time; this presents a challenge to the investigator. Here, the authors have shown that there is a statistically significant overexpression of CDK2AP1 associated with overall survival and cause-specific survival for specific molecular subtypes of prostate cancer (Pca) in the clinical setting. Of particular interest is the evidence for a direct p53-CDK2AP1 interaction (further studies will be needed to delineate the mechanism and consequences of an interaction).

I believe the diversity of PCa patient population is off set by the large cohort (n = 275); the findings will be interest to the pathology community, oncologists, and the wider community of those with research interests in cancer biomarker validation, beyond PCa.

The Materials and Methods are largely standard, relevant to the study objectives, appropriate and described clearly and adequately in detail.

The results of the study are for the most part presented well, and the interpretations are supported by evidence-based data, without overinterpretation. The statistical treatment of the data is sound.

There is an issue beginning on page 10, line 286. This refers the reader to Supplementary tables 2 and 3. However, Supplementary Table 2 (contents not re docking study) page 16 of the manuscript is an exact repeat of Table 1. Obviously, this requires correction. Furthermore, there is no Supplementary Table 3 in the manuscript. Consequently, the supporting information relating to the molecular modelling is not presented. No doubt the data exists but given the importance of the finding on the p53- CDK2AP1 interaction, this should be rectified as a priority.

Response: The supplementary tables were corrected and missed Supplementary. Tables 2, 3 were added as Supplementary 3 and 4

In addition, it would be helpful to state the number of iterations that were applied in the in-silico derivation of the energy-minimised modelled structure of the p53 – CDK2AP1 interaction.

Response: From the in-silico derivation, it was observed that CDK2AP1 exhibited 15 hydrogen bonding contacts with p53 residues. Additionally, the modelled interactions also demonstrated 29 physical contacts within 4 angstroms of interacting p53 residues (Supplementary tables 3 and 4).

It is recommended that the docking diagram Figure 3 (C) is displayed physically larger in the manuscript; this would be normal practice and will allow for identification of key amino acid residues (indicated by arrows) that are regarded as the key, strongest interactions.

Response:  The figure 3C were enlarged as recommended in the revised manuscript.

General comment. Although abbreviations are defined immediately after they appear in the text of the manuscript, it is recommended that a separate abbreviation list is included (subject to editorial policy); this should be helpful to those non-PCa specialists unfamiliar with some terms, either new to the area or are part of the wider biomarker research community to whom this paper will be of interest.

Response: the abbreviations list was added as recommended

My recommendation is that the article merits publication subject to amendments outlined above.

Round 2

Reviewer 1 Report

thank you for the revised manuscript